# Direct-Acting Antiviral Use for Genotype 1b Hepatitis C Patients with Associated Hematological Disorders from Romania

**DOI:** 10.3390/medicina57090986

**Published:** 2021-09-18

**Authors:** Iosif Marincu, Felix Bratosin, Manuela Curescu, Oana Suciu, Mirela Turaiche, Bianca Cerbu, Iulia Vidican

**Affiliations:** Methodological and Infectious Diseases Research Center, Department of Infectious Diseases, “Victor Babes” University of Medicine and Pharmacy, 300041 Timisoara, Romania; imarincu@umft.ro (I.M.); manuela.curescu@gmail.com (M.C.); suciu.oana@umft.ro (O.S.); mirela.paliu@gmail.com (M.T.); ionitabiancaelena@yahoo.com (B.C.); iulia.georgianabogdan@gmail.com (I.V.)

**Keywords:** hepatitis C, direct-acting antiviral therapy, hematological disorders, Viekirax, Exviera, Harvoni

## Abstract

*Background and objectives*: this study assessed variations in the blood parameters of patients with hematological disorders infected with HCV throughout a 12-week interferon-free treatment regimen. *Materials and methods:* We followed a total of 344 patients suffering from chronic hepatitis C, infected with the 1b genotype and concomitant hematological disorders, who benefited from the direct-acting antiviral (DAA) therapy in our clinic. Seven of the most routinely checked blood parameters were analyzed, namely, hemoglobin, leucocyte count, neutrophils, erythrocyte count, platelet count, ALT, and total bilirubin level. In total, 129 patients received a treatment scheme comprising ombitasvir, paritaprevir, ritonavir, and dasabuvir, while the 215 other patients received a sofosbuvir and ledipasvir regimen. *Results:* Patients enrolled in the study showed remarkably increased ALT levels in the first four weeks of DAA treatment, normalizing to levels below 40 U/L by the end of regimen. There were no other blood parameters that worsened throughout the 12-week regimen to levels below our laboratory’s normal range. After 12 weeks of DAA therapy, 309 patients (90%) achieved SVR. *Conclusions:* Our findings are consistent in evaluating the efficacy and tolerability of direct-acting antivirals for 1b genotype HCV infected patients with associated hematological malignancies under remission, and other hematological disturbances, that were previously unsuccessfully treated with a pegylated interferon regimen. Thus, paving a pathway for government-funded programs being implemented in this direction.

## 1. Introduction

Chronic hepatitis C is a liver infection caused by the hepatitis C virus (HCV), an enveloped, single-stranded linear ribonucleic acid (RNA) virus of the Flaviviridae family with six major genotypes. About 180 million people globally are chronically infected with the HCV [1]. According to a 2018 study [2], more than 380,000 Romanians suffer from chronic hepatitis C (CHC), placing the country among others in the top of the ranking list of European countries with the highest numbers of patients with CHC. The genotype 1b of the HCV is endemic to the European region, and the most prevalent in Romania [3]. The prevalence of genotype 1b in the general population reaches 4.9%, while in Romania, it is responsible for more than 50% of all HCV infections [4]. Chronic HCV is characterized by the persistence of HCV RNA in the blood for over six months after acute infection [5]. Around 54% to 86% of patients diagnosed with acute HCV infection develop a chronic illness [6]. With the disease’s progression, at least 20% of chronically infected individuals develop liver fibrosis and cirrhosis [7], while being at risk for life-threatening complications such as hepatocellular carcinoma and end-stage liver disease [8].

The liver plays a vital role in the storage of folic acid, iron, and vitamin B12, and the production of inhibitors and clotting factors. Thus, progressive destruction leads to modifications in hematological parameters that are dependent on these molecules. The most common blood abnormalities seen in patients with chronic HCV infections are thrombocytopenia, anemia, leukopenia, and neutropenia. These complications can influence HCV treatment and adherence, which could compromise outcomes. Thrombocytopenia is defined as a decrease in platelet count below the lower standard limit (i.e., <150 × 10^9^/L). The thrombocytopenia mechanism is either triggered by reduced thrombopoietin production by the liver or having an autoimmune component, resulting in autoimmune thrombocytopenia and splenic sequestration due to hypersplenism [9]. Patients with advanced chronic liver disease frequently experience thrombocytopenia, increasing incidence correlating with the severity of liver disease [10].

Over the last ten years, the therapeutic options for patients with HCV have evolved, being revolutionized by the development of direct-acting antiviral agents. Until 2015, the standard treatment for CHC was represented by the combination of pegylated interferon and ribavirin. Studies showed that anemia is a common complication of either single or combination therapy [11], defined by a decrease in hemoglobin levels ≤12 g/dL. Clinical cases showed that, besides treatment-induced anemia, patients with hepatitis C infection could develop autoimmune hemolytic anemia in the absence of treatment with pegylated interferon and ribavirin [12]. While the interferon-based regimen for CHC infection had low rates of success [13] and high treatment discontinuation rates due to side effects, the novel interferon-free anti-HCV specifically targeted therapy is available with higher sustained virological response (SVR) rates and is more tolerable to the patient [14].

Although the newest DAA treatment regimens are proved to outperform pegylated interferon and ribavirin in terms of safety and side effects, in Romania the use of DAA in hematologically-ill patients has been seen still as a clinical problem, physicians being reluctant in administering this medication to this group of patients. This led us to evaluate changes in serum parameters associated with DAAs during the course of a 12-week regimen, having the main purpose in determining whether the patients with hematological malignancies under remission and other hematological disorders remain stable during the DAA treatment, and achieve SVR after previously being unsuccessfully treated with a pegylated interferon regimen for CHC infection.

## 2. Materials and Methods

This research follows an observational study design, detailing our prospective cohort, evaluating data collected by the First Department of Infectious Diseases at the Dr. Victor Babes Clinical Hospital for Infectious Diseases and Pulmonology in Timisoara, Romania. The study comprised patients diagnosed with CHC who had received interferon-free therapy from 1 January 2015 to 31 December 2018. Hepatitis C was diagnosed according to currently available guidelines [15], assessing the HCV RNA viral load by reverse-transcription PCR. Patients with quantitative HCV RNA results <15 IU/L were excluded from commencing the direct-acting antiviral treatment.

Previously to benefiting from the new direct-acting antiviral treatment, the patients included in the current study had previously been unsuccessfully treated with interferon-based regimen Pegasys^®®^ + Copegus^®®^, available in Romania. All patients enrolled in our study underwent a standard 12-week treatment regimen, being monitored before treatment initiation and at every four weeks for any unusual changes in their blood patterns. Across the first two years of the study, patients received a treatment scheme consisting of Viekirax^®®^ (a combination of ombitasvir, paritaprevir, and ritonavir) and Exviera^®®^ (dasabuvir). In the two following years, patients received the Harvoni^®®^ regimen (a combination of sofosbuvir and ledipasvir).

During a 4-year period, a total of 680 CHC patients with oncohematological disease in remissionwere initially identified. After excluding noneligible cases, 344 patients accepted to enroll in the study (Figure 1). The hematological disorders included in our study comprised leukemias and lymphomas under remission, multiple myeloma in remission, hemophilia under treatment, and previous anemia, while noneligible cases were patients in their acute phase of leukemia or lymphoma, and those undergoing induction therapy due to drug–drug interaction concerns. Patients who did not have the 1b genotype were also excluded from the study, regarding a possible different response to the direct-acting antiviral treatment, as some studies suggest [16], and patients coinfected with a different hepatitis C virus genotype [17]. Another exclusion criterion was patients associated with a coinfection with another hepatitis virus (A, B, D or E). Uncompliant patients and those who had died during treatment due to unrelated causes were also excluded from our data collection. Eventually, chronic kidney disease patients were not excluded from the study, considering research [18] being conducted on direct-acting antiviral treatment for CHC patients showing no safety concerns.

Data collected comprised demographic information (age, gender, height, weight, and body mass index), blood test results (hemoglobin, leucocyte count, neutrophils, erythrocyte count, thrombocytes, alanine aminotransferase, and total bilirubin), and the treatment scheme and date of the blood test. The liver fibrosis stage based on ultrasound elastography was established for all patients and classified as: (i) absent or mild fibrosis, (ii) significant fibrosis, (iii) severe fibrosis, or (iv) cirrhosis [19].

All selected patients were subjected to the following: history, clinical examination, and laboratory tests. Whole venous blood was drawn into EDTA anticoagulated tubes and immediately sent to the laboratory at ambient temperature. Blood analysis was performed using an automated hematology analyzer (Sysmex XN 1000). Serum HCV RNA was measured utilizing the COBAS AmpliPrep/COBAS TaqMan HCV Quantitative Test, version 2.0 (Roche Molecular Systems).

Data were analyzed using IBM SPSS version 26.0.0.0 for Windows. A normality test was conducted before performing one-way ANOVA to assess differences at every four weeks, and two-factor ANOVA assessed particular differences by factoring the dependent variables on the basis of the dates of the blood test and treatment scheme. A χ^2^ test was applied to compare proportions between genders. Lastly, multivariate analysis was performed, and linear regression was run to observe how the treatment contributed in time to the observed changes in blood parameters.

The Local Commission of Ethics for Scientific Research from the Victor Babes Clinical Hospital for Infectious Diseases and Pulmonology in Timisoara operates under article provisions 167 of Law no. 95/2006, art. 28, Chapter VIII of Order 904/2006, with EU GCP Directives 2005/28/EC, International Conference of the Harmonisation of Technical Requirements for Registration of Pharmaceuticals for Human Use (ICH), and with the Declaration of Helsinki Recommendations Guiding Medical Doctors in Biomedical Research Involving Human Subjects. The current study was approved on 10 January 2016 with approval number 7792. All study participants agreed to be involved in this study by signing an informed consent form.

## 3. Results

From a total of 344 (100%) participants in the study, 129 (37.5%) patients received the ombitasvir, paritaprevir, ritonavir, and dasabuvir treatment scheme, while the remaining 215 (62.5%) patients received the sofosbuvir and ledipasvir regimen. Our sample included 39 (11.3%) patients with acute myeloid leukemia, 8 (2.3%) patients with acute lymphocytic leukemia, 51 (14.8%) patients with chronic lymphocytic or myeloid leukemia, 13 (3.7%) cases of multiple myeloma, 104 (30.2%) cases of iron and vitamin deficiency anemias, 12 (3.4%) patients suffering from sickle-cell anemia, 88 (25.5%) patients with anemia associated with malignancies other than leukemia and lymphoma, and 29 (8.4%) patients with hemophilia. Data were normally distributed for all variables involving hematological changes. At the same time, except for platelet count (*p* value < 0.0000) and alanine aminotransferase (*p* value < 0.0000), the comparison between the two treatment groups (Table 1) did not show statistically significant differences. Thus, analysis was continued without stratification by treatment scheme. In the same manner, proportions between genders did not return any significant differences (*p* value = 0.5336).

One-way ANOVA by date of treatment (Table 2) showed statistically significant changes during the 12-week regimen in hemoglobin (*p* value < 0.0000), neutrophils (*p* value < 0.0000), erythrocyte count (*p* value 0.0312), ALT (*p* value < 0.0000), and total bilirubin (*p* value < 0.0000). In the same manner, post hoc analysis returned significant changes every four weeks, predominantly in hemoglobin level, neutrophils, ALA, and total bilirubin level. However, the findings did not change to pathological values compared to our laboratory’s normal range. None of the tested blood parameters worsened outside physiological levels throughout the 12-week regimen, even though there were statistically significant changes at every four weeks, and between the start and finish of the DAA regimen. There was also no significant difference in liver fibrosis stages between the two study groups, although there was an alarming percentage of liver cirrhosis cases (18.6% vs. 19.0%).

Linear-regression analysis (Table 3) showed a statistically significant impact of the DAA therapy, measured in terms of the time since the patients commenced the treatment, on hemoglobin levels (CI: −0.069 to −0.025, *p* value < 0.000), neutrophils (CI: 0.119–0.350, *p* value < 0.000), total bilirubin (CI: 0.019–0.036, *p* value < 0.000), and ALT (CI: −3.735 to −2.989, *p* value < 0.000), the last having the only significant effect size of change. Hemoglobin and ALT had negative trends during treatment (OR: −0.111, *r*^2^ = 0.012), and (OR: −0.431, *r*^2^ = 0.185) respectively. Neutrophils and total bilirubin had positive trends during treatment (OR: 0.107, *r*^2^ = 0.011; and OR: 0.172, *r*^2^ = 0.030, respectively).

As shown in the statistical findings and Figure 2, patients started the DAA regimen with generally good hematological parameters, with average values within the normal range except for ALT. Patients commenced the new treatment with an average ALT level around 72 U/L. Still, their hepatic function drastically improved in the first four weeks of the treatment, normalizing the ALT to levels below 40 U/L, and finishing the regimen on the same level. At 12 weeks or more after the treatment regimen onset, all 344 patients (100%) who followed the full treatment course were confirmed with an HCV RNA viral load (<25 IU/mL), indicating a sustained virologic response (SVR) [20], while 309 patients (90%) had an undetectable viral load.

## 4. Discussion

This study evaluated the effect of DAA (ombitasvir, paritaprevir, ritonavir, dasabuvir, sofosbuvir, and ledipasvir) therapy on patients with CHC by tracking hematological changes during a 12-week treatment with DAAs. Our results agree with those of Feld et al. [21], who found that hematologic abnormalities were infrequent in patients treated with sofosbuvir and velpatasvir, affecting 1% of patients or less. Although our group of patients was not treated only with sofosbuvir and velpatasvir, we did not find any significant differences in patients’ serum parameters treated with ombitasvir, paritaprevir, ritonavir, dasabuvir, sofosbuvir, and ledipasvir.

A multicentric study developed in Poland [22] included 209 patients to assess the DAA’s efficacy in patients taking concomitant treatment. All patients were treated with the ombitasvir, paritaprevir, ritonavir, and dasabuvir formula, while at the end of the study, SVR was achieved at a minimal rate of 95.1%, although during treatment, 15.8% of the patients required treatment modification due to drug-to-drug interaction. Considering that our study did not test any drug-to-drug interactions or any treatment changes during the curative weeks, our results are consistent with those of the study of Krzysztof et al., with more than 90% of the patients achieving SVR.

In another study conducted by El-Kholy Am et al. [23], the results showed that the platelet count initially decreased after starting the treatment with DAA, increased gradually, and went back to the initial normal values at the end of the 12th week. In our study, we observed the same changes, with patients’ hematological parameters showing a dip in platelet count, and in erythrocyte count and hemoglobin in the first four weeks from starting the treatment, which is consistent with the study of El-Kholy Am et al.

We excluded from the current study patients undergoing induction treatment for leukemia and lymphoma due to drug–drug interaction concerns, although several studies [24] report the safety of concomitant DAA therapy and chemotherapy for different malignancies.

For eligibility purposes and to avoid biased results by including all HCV genotypes, our study chose the 1b genotype, this being the most prevalent in Romania. However, clinical trials [25] showed that SVR rates are exceeding 90% irrespective of genotype, using the same treatment scheme as our study, with ombitasvir, paritaprevir, ritonavir, and dasabuvir. Moreover, meta-analysis on the same treatment scheme, including diverse populations, showed SVR ranging from 91% to 100% at 12 weeks of treatment, making our study slightly underachieving regarding our SVR rate and population homogeneity.

A limitation of our prospective study was that patients included in the first half, during 2016–2018, had been treated with an interferon-based regimen before starting DAA therapy. This probably made them more likely to enroll in the study with worse blood parameters and general health condition due to the longer exposure to HCV than those of the second group, treated from 2018 to 2020 with the sofosbuvir and ledipasvir regimen.

Another possible limitation of the current study is the number of blood parameters that were assessed. Other, more specific analyses were inconsistently checked during the 4-year period, raising the question of which other parameters might have changed during the course of the 12-week treatment scheme.

In this paper, our findings strengthened the results of previous studies and clinical trials for the treatment of chronic HCV infection with two treatment schemes (ombitasvir, paritaprevir, and ritonavir, associated with dasabuvir; and sofosbuvir, associated with ledipasvir), showing, once again, their efficacy either for patients where previous interferon regimens had failed or for those recently infected who did not undergo any treatment. Proving on a medium-scale sample of 344 chronic HCV patients infected with the 1b genotype suffering from concomitant hematological disturbances that the main blood parameters mainly improve, leading to a more than 90% achievement of SVR, the study opens the territory for large-scale government-funded programs aiming to cure one of the most severely affected countries in Europe from chronic HCV.

## 5. Conclusions

Current direct-acting antivirals have proven, generally and in our study, to be very effective in treating the HCV infection. Over 90% of the patients with hematological malignancies under remission, and hematological disorders included in this prospective study have achieved SVR by using two different treatment regimens using the association of ombitasvir, paritaprevir, ritonavir, and dasabuvir, and the single-therapy scheme with sofosbuvir and ledipasvir association. The above-mentioned treatment regimens are not significantly different in achieving the desired results. They do not seem to worsen the patients’ condition throughout the treatment, making them very tolerable and effective for patients infected with Type 1b HCV compared to the old interferon-based therapies.

## Figures and Tables

**Figure 1 medicina-57-00986-f001:**
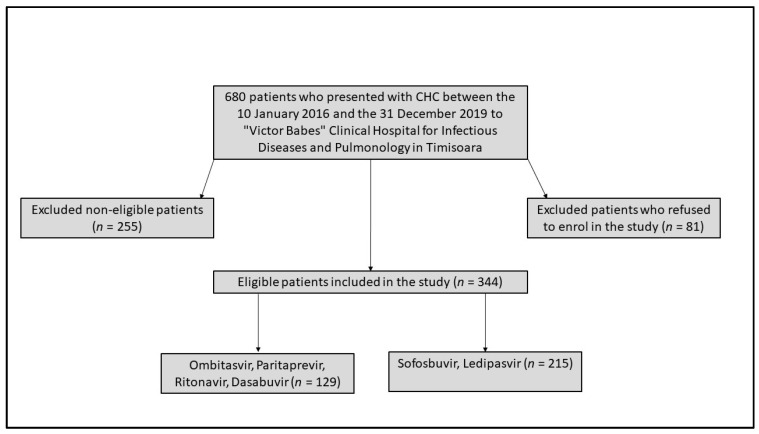
Flowchart of patient recruitment.

**Figure 2 medicina-57-00986-f002:**
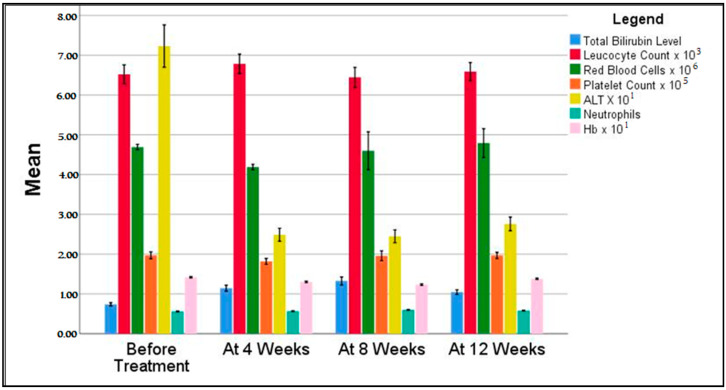
Evolution of blood tests during treatment. Average laboratory values of patients involved in the study were evaluated in a follow-up scheme, with a total of 4 measurements. Hb = Hemoglobin; ALT = Alanine Aminotransferase.

**Table 1 medicina-57-00986-t001:** Study sample’s general characteristics, with demographics and average laboratory values recorded from the study participants. Data in this table were collected at 12 weeks, split by treatment options, and compare means and proportions between the two study groups.

	Subtypes/Range	Number (%) or Mean (SD)		*p* Value
		Group 1 * (*n* = 129)	Group 2 ** (*n* = 215)	
Gender	Male	45 (34.9)	68 (31.6)	0.5336
	Female	84 (65.1)	147 (68.3)	
Age (years)		60 (7.4)	59 (11)	0.0684
Weight (kg)		77 (13.7)	74 (15.3)	0.0004
Height (cm)		165 (8.4)	165 (9.1)	0.8485
BMI		28 (4.4)	27 (4.7)	<0.0000
Hemoglobin (g/dL)	14–18 g/dL (male) 12–16 (female)	13.2 (1.5)	13.3 (1.7)	0.4298
Leucocyte count (x/mm^3^)	4.000–8.000/mm^3^	6292 (2329)	6758 (2115)	0.0662
Neutrophils (%)	55–70%	57.8 (10.36)	57.6 (9.6)	0.7097
Erythrocyte count (x/mm^3^)	4.5–6 million/mm^3^	4,555,509 (549,104)	4,574,418 (688,314)	0.9100
Platelet count (x/mm^3^)	150.000–400.000/mm^3^	173,327 (70,225)	204,541 (76,978)	<0.0000
ALT (U/L)	10–40 U/L (male) 10–34 U/L (female)	42.64 (54.7)	34.1 (44.8)	<0.0000
Total bilirubin (mg/dL)	0.1–1.2 mg/dL	1.13 (0.41)	1.01 (0.28)	0.0520
Liver fibrosis stage				0.0842
	Absent or mild	69 (53.3)	91 (42.3)	
	Significant	22 (17.0)	39 (18.1)	
	Severe	14 (10.8)	44 (20.4)	
	Cirrhosis	24 (18.6)	41 (19.0)	

* Ombitasvir, paritaprevir, ritonavir, and dasabuvir. ** Sofosbuvir and ledipasvir. BMI = Body Mass Index. ALT = Alanine Aminotransferase.

**Table 2 medicina-57-00986-t002:** Average values of blood tests during treatment comparing main hematological parameters evaluated by the average results obtained before treatment and the follow-up period. All data were compared by ANOVA between groups and by a multivariate method.

	Before Treatment *		At 4 Weeks		At 8 Weeks		At 12 Weeks		Two-Way ANOVA	Multivariate
	Group 1	Group 2	Group 1	Group 2	Group 1	Group 2	Group 1	Group 2	*p* Value	*p* Value
Hemoglobin	14.3	14.1	12.6	13.3	12.4	12.3	13.9	13.7	<0.0000	0.0010
Leucocyte count	6153	6738	6727	6816	6394	6472	5894	7005	0.2320	0.0090
Neutrophils	55.4	56.5	58.7	55.6	60.4	59.8	57.0	56.8	<0.0000	0.0068
Erythrocyte count	4,637,054	4,723,209	4,030,023	4,282,558	4,242,325	4,812,000	5,312,635	4,479,907	0.0312	0.0131
Platelet count	153,085	224,065	180,364	182,646	182,779	198,740	170,375	212,714	0.0673	0.0673
ALT	86.5	63.7	27.8	23.1	24.1	24.7	32.1	24.9	<0.0000	<0.0000
Total bilirubin	0.97	0.60	1.3	1.1	1.3	1.3	1.0	1.1	<0.0000	<0.0000

* Data collected the first day before commencing treatment. ALT = Alanine Aminotransferase. ANOVA = Analysis of Variance.

**Table 3 medicina-57-00986-t003:** Regression analysis presenting odds ratios with confidence intervals and the level of significance when testing to estimate the effect that treatment duration had on dependent variables (hematological parameters).

Independent	Blood Test	OR	95% CI	*p* Value	*r* ^2^
Date of Blood Test	Hemoglobin	−0.111	−0.069 to −0.025	<0.000	0.012
	Leucocyte count	−0.007	−30.030 to 23.417	0.808	0.001
	Neutrophils	0.107	0.119–0.350	<0.000	0.011
	Erythrocyte count	0.028	−16.00 to 51.71	0.301	0.001
	Platelet count	0.017	−7.036 to 13.95	0.518	0.001
	ALT	−0.431	−3.735 to −2.989	<0.000	0.185
	Total bilirubin	0.172	0.019–0.036	<0.000	0.030

ALT = Alanine Aminotransferase.

## Data Availability

Data available on request.

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
