# Peer review of "Direct-Acting Antiviral Use for Genotype 1b Hepatitis C Patients with Associated Hematological Disorders from Romania"

_medicina, 2021, doi:10.3390/medicina57090986_

Round 1

Reviewer 1 Report

In the manuscript “The Results of Direct-Acting Antivirals use for Genotype 1b Hepatitis C Patients with Associated Hematological Disorders from Romania”, Bianca Cerbu and co-authors aimed to assess the variations in blood parameters of patients with hematological disorders infected with HCV throughout their 12-week interferon-free treatment. In a total of 344 patients (129 with ombitasvir, paritaprevir, ritonavir, and dasabuvir and 215 with sofosbuvir and ledipasvir), no blood parameters, but not ALT level, worsened throughout the 12-week regimen and 90% of patients had an undetectable viral load. It sounds good when the authors concluded that their findings are consistent in evaluating the efficacy and tolerability of direct-acting antivirals for HVC infections of 1b genotype patients from Romania. However, this manuscript should be improved by following concerns.

  1. Conclusion is too long. It should reflect upon the aims.
  2. What is the originality of present study? The increase of SVR rate up to >90% and blood parameter normalization were already reported as authors discussed. In addition, all blood parameters, except ALT level, before treatment are in normal range.
  3. Figure and Tables should have a short explanatory caption, unit to help readers easy to follow your work.
  4. Data presented in Table 1, Table 2 Column 1 and Figure 1 Group 1 are all at baseline? The authors should describe the time point that data assessed in Table 1.
  5. SVR means that the hepatitis C virus is not detected in the blood 12 weeks or more after completing treatment.In present study, it should state that “not detected at the end of treatment”.
  6. The parts M&M and Results should have sub-heading but not the part Discussion. Measuring HCV RNA (pages 76-77 and 113-114) should be described in one paragraph.
  7. In the abstract, line 19 “… ALT levels, averaging 70 U/L while DAA therapy was ongoing, but returned to the normal range at the end of the treatment” should be change to “remarkedly decreased in the first four weeks of the treatment, normalizing to levels below … U/L till the end of regimen”. Number should be added in line 192, below … U/L.
  8. List of references must be prepared following the Instructions for Authors.
  9. Ombitasvir, paritaprevir, ritonavir, and dasabuvir, sofosbuvir and ledipasvir can be abbreviated or grouped into regimens 1 and 2.

Author Response

Dear reviewer team,

Thank you for considering our manuscript, as we appreciate your efforts to analyze and improve our paper. Thus, based on the feedback received during the review phase, our team had carefully revised the article with the following changes:

  1. The conclusion part was shortened to reflect upon aims and results only.
  2. We believe the originality of the present study is represented by the unique cohort of patients from Romania. The vast majority are of Romanian ethnicity, they all share the type 1b HCV genotype, and they were all known to have a hematologic comorbidity at the time of treatment.
  3. An explanatory caption was added for figures and tables.
  4. Line 73 was deleted
  5. Line 83 explains when was the time point for data collection.
  6. Line 96: We have included a flowchart of patients selection. We did not control the patients’ blood exams before the DAA therapy as inclusion criteria.
  7. Line 101: We specify that selected cases of leukemia and lymphoma were in the remission phase when the DAA treatment was given.
  8. Line 102: Anemia of all causes was referred as generally low levels of erythrocytes or hemoglobin.
  9. Lines 127-140 We couldn’t fit the normal range in table 1, but modified this paragraph in the materials and methods.
  10. Line 205 was modified to explain the meaning of SVR in our study.
  11. The subsections in the discussion part were removed, as suggested.
  12. Lines 19 and 192 were modified to describe the observed ALT values.
  13. The list of references was properly formatted following the Instructions for Authors guidelines.
  14. Ombitasvir, paritaprevir, ritonavir, and dasabuvir, sofosbuvir and ledipasvir were abbreviated as Group 1 and 2.

Best regards,

The authors

Reviewer 2 Report

  • the most important point to explain very well is the type of selected patients: all the pts are not in treatment? All the pts had normal blood exams before the therapy with DAA? What means "anemia of all causes"? The pts were treated with DAA when the hematological disease was in remission?
  • Revised title
  • Revised all tables: delete lines 116-125 and put normal range in table 1; improve layout 
  • Delete line 73 " including....duration"

Author Response

Dear reviewer,

Thank you for considering our manuscript, as we appreciate your efforts to analyze and improve our paper. Thus, based on the feedback received during the review phase, our team had carefully revised the article with the following changes:

  1. The conclusion part was shortened to reflect upon aims and results only.
  2. We believe the originality of the present study is represented by the unique cohort of patients from Romania. The vast majority are of Romanian ethnicity, they all share the type 1b HCV genotype, and they were all known to have a hematologic comorbidity at the time of treatment.
  3. An explanatory caption was added for figures and tables.
  4. Line 73 was deleted
  5. Line 83 explains when was the time point for data collection.
  6. Line 96: We have included a flowchart of patients selection. We did not control the patients’ blood exams before the DAA therapy as inclusion criteria.
  7. Line 101: We specify that selected cases of leukemia and lymphoma were in the remission phase when the DAA treatment was given.
  8. Line 102: Anemia of all causes was referred as generally low levels of erythrocytes or hemoglobin.
  9. Lines 127-140 We couldn’t fit the normal range in table 1, but modified this paragraph in the materials and methods.
  10. Line 205 was modified to explain the meaning of SVR in our study.
  11. The subsections in the discussion part were removed, as suggested.
  12. Lines 19 and 192 were modified to describe the observed ALT values.
  13. The list of references was properly formatted following the Instructions for Authors guidelines.
  14. Ombitasvir, paritaprevir, ritonavir, and dasabuvir, sofosbuvir and ledipasvir were abbreviated as Group 1 and 2.

Best regards,

The authors

Round 2

Reviewer 1 Report

Manuscript is much better. One remaining concern is Odds Ratio in Table 3. Negative trends mean that the odds ratio is smaller than 1.

Author Response

Dear reviewer,

Thank you once again for helping us improve our manuscript. We believe all concerns from the first round of review were accurately solved, and we continued with several edits based on the secondary feedback, as follows:

  1. Indeed, a negative OR is smaller than 1, although our significant findings with negative trends (hemoglobin levels, and ALT) din not include the values of 1 or 0. However, ALT makes for the only variable with a significant effect size, since its Confidence Interval is smaller than -1, while the CI for hemoglobin levels is between 0 and -1. The same rule applies for the other significant findings that have a positive CI between 0 and 1, but not including the margins.
  2. We have added a short paragraph in the introduction part, underlying the main purpose of the study.
  3. We have tried to improve the tables and figures as much as possible.
  4. The paper was checked by MDPI English editing service, and was modified accordingly.
  5. Overall, we believe the paper was greatly improved from the initial version.

Best regards,

The authors

Reviewer 2 Report

  • The are not substancial modification, in particular title is not modified and tables are not improved
  • It is known that DAA is allowed also during or after chemotherapy for hematological tumor without more toxicity. In this paper the most important limitation remains the type of selected patients: are all in hematological remission?
  • Unfortunatly the available laboratory exams are very limited and it is quite impossible to understand the improvement of liver function only with ALT and bilirubin. 

Author Response

Dear reviewer,

Thank you once again for helping us improve our manuscript. We believe all concerns from the first round of review were accurately solved, and we continued with several edits based on the secondary feedback, as follows:

  1. Indeed, concurrent use of DAAs with antineoplastic agents can be performed within safe limits, although only with selected antineoplastic agents under close monitoring for drug-drug interactions. At the time of study commencement in 2015, the DAAs were quite a new spectrum of medications in Romania, and hematologists were reluctant in administering them along with antineoplastic agents. The same thing still happens to many cases in our region.
  2. Our main purpose was not to investigate if the liver function tests normalize during and after treatment, but particularly to determine whether the patients with hematological disorders under remission remain stable during the DAA treatment. It is important to mention that all these patients were previously unsuccessfully treated with pegylated interferon. We should also underline that our cohort comprised approximately 30% of patients with severe liver fibrosis or cirrhosis.
  3. We have tried to improve the tables and figures as much as possible.
  4. We have added a short paragraph in the introduction part, underlying the main purpose of the study.
  5. The paper was checked by MDPI English editing service, and was modified accordingly.
  6. Overall, we believe the paper was greatly improved from the initial version.

Best regards,

The authors

Round 3

Reviewer 2 Report

  • Revised title: delete "Results of"
  • Put always "HCV", after line 32
  • Put always "CHC" after line 32
  • Re-write lines 35 to 40; lines 52 to 56
  • Put "(DAA)" in line 60
  • Add into introduction a sentence in which you explain as in Romania the use of DAA in hematological patients has been seen still as a clinical problem
  • Delete " at the....in Timisoara" in lines 100-101
  • Specify lines 103-104: "multiple myeloma under remission"; hemophilia in treatment?; "previous" anemia? Should be possible describe these selected patients as "oncohematological disease in remission" and "not oncohematological disease in remission"? 
  • Lines 127-131: put all normal range in table 1 and delete these lines
  • Delete lines 132-134
  • Specify "chronic leukemia" in line 158: lymphocitic? Myeloid? 
  • Table 2 is not correctly formatted 
  • Delete lines 257-260 "here....level"
  • Revised all references using the correct form 

Author Response

Dear Reviewer,

Thank you for your efforts and the helpful feedback. Please check below the following changes:

  1. The authors’ order was changed, and we all signed the authorship change form.
  2. Title was revised
  3. HCV and CHC are now used all over the text
  4. Lines 35 to 40: Following a research conducted in 2018 [2], it was discovered that more than 380,000 Romanians have Chronic Hepatitis C (CHC), putting the nation among the top European countries in terms of numbers of people suffering from the disease. The genotype 1b of the hepatitis C virus (HCV) is endemic to the European area, and it is the most common in Romania [3]. Across the globe, genotype 1b infects approximately 5% of the total population with CHC, but in Romania it is responsible for more than half of all HCV infections [4].
  5. Lines 52 to 56: The mechanism of thrombocytopenia is either due to decreased thrombopoietin production by the liver or has an autoimmune component, resulting in autoimmune thrombocytopenia due to splenic sequestration [9]. As a result, patients with advanced chronic liver disease frequently experience thrombocytopenia, with the incidence increasing proportionally to the severity of the disease [10].
  6. We included “in Romania the use of DAA in hematological patients has been seen still as a clinical problem” in the last paragraph from introduction.
  7. Deleted “at the… in Timisoara” from lines 100-101, and added “patients with oncohematological disease in remission”
  8. Revised lines 106-108
  9. Lines 127-131 were deleted and added to table 1 as a column with normal ranges.
  10. Deleted lines 132-134
  11. Line 162: added chronic lymphocytic or myeloid leukemia
  12. Formatted Table 2
  13. Deleted lines 261-264 (previously 257-260): here…level
  14. We revised all references using the correct form.

Best regards,

The authors
